# Fungal Abscess of Anterior Nasal Septum Complicating Maxillary Sinus Fungal Ball Rhinosinusitis Caused by *Aspergillus flavus*: Case Report and Review of Literature

**DOI:** 10.3390/jof10070497

**Published:** 2024-07-18

**Authors:** Shih-Wei Yang, Cheng-Ming Luo, Tzu-Chien Cheng

**Affiliations:** 1Department of Otolaryngology-Head and Neck Surgery, Chang Gung Memorial Hospital, Keelung 204, Taiwan; lcm3647@cgmh.org.tw; 2College of Medicine, Chang Gung University, Taoyuan 333, Taiwan; 3Department of Pathology, Chang Gung Memorial Hospital, Keelung 204, Taiwan; tbeitb@cgmh.org.tw

**Keywords:** fungal abscess, anterior nasal septum, fungal ball, mycetoam, diabetes mellitus, fungus, *Aspergillus flavus*

## Abstract

Anterior nasal septum abscess is not a rare clinical disease entity. In terms of the etiologies of the disease, bacteria are obviously more common than fungi. Fungal culture and pathological examination are essential for diagnosis of a fungal abscess of the anterior nasal septum and the basis of prescription of antifungal agents. We report a 57-year-old male patient who came to our outpatient clinic due to refractory nasal congestion for 3 weeks despite receiving treatments by a local medical doctor. Radical surgery with postoperative adjuvant radiotherapy for the right buccal cancer was carried out 14 years ago. The patient has diabetes mellitus and the blood sugar level has been well controlled by oral hypoglycemic agents over the past several years. Computed tomography revealed an abscess in the anterior septum along with rhinosinusitis. Incision and drainage of the nasal septum abscess and functional endoscopic sinus surgery were carried out. Fungal culture and pathological examination confirmed a fungal abscess in the anterior nasal septum and fungal ball rhinosinusitis. Antibiotics and an antifungal agent were given, and the postoperative course was uneventful. A dialectical argument was made regarding the causal relationship between the fungal abscess of the anterior nasal septum and maxillary fungal ball sinusitis. A literature review of the previous case reports was carried out to elucidate the immune status of patients of this disease. In order to reach a rapid establishment of a fungal abscess of the anterior nasal septum, clinicians should keep this disease in mind and remain vigilant. An immuno-compromised status is more commonly found in patients with fungal abscess of the anterior nasal septum and is another important characteristic of this disease. Prompt diagnosis and effective treatment are equally important in patients with lower immune status of this kind, and the latter is based on the former.

## 1. Introduction

A fungal abscess is not a common cause of anterior nasal septum abscesses. It is a rare occurrence, with bacterial infections, particularly Staphylococcus aureus, being more common culprits [1]. Fungal infections of the nasal cavity typically occur in immunocompromised individuals and are known for their destructive properties. However, fungal infection as the sole cause of an anterior nasal septum abscess is rare, with only a few cases documented in the literature [1,2,3,4,5,6,7,8,9,10,11]. Consequently, when clinicians encounter an anterior nasal septum abscess, they often initially prescribe empiric antibiotics, including beta-lactamase-resistant antibiotics, which are generally effective. As for a fungal abscess of the anterior nasal septum, empiric antibiotics usually do not work. In such cases, fungal culture and pathological examination play a crucial role in establishing an accurate diagnosis [1,3,4,6,12,13,14,15].

We describe the case of a patient who received radical surgery and postoperative radiotherapy for right buccal cancer 14 years ago, has a history of well-controlled type 2 diabetes mellitus, hypertension and mixed hyperlipidemia, and suffered from an anterior nasal septum abscess, which was refractory to aspiration drainage of the abscess and oral beta-lactamase antibiotic treatment. Preoperative computed tomography revealed a concomitant anterior nasal septum abscess and sinusitis. Postoperative fungal culture and pathological examination showed that the etiology of the nasal septum abscess and sinusitis was *Aspergillus*. The patient was successfully treated with surgical drainage of the abscess, functional endoscopic sinus surgery and postoperative antifungal therapy.

## 2. Case Presentation

A 57-year-old male did not have any nose discomfort until 3 weeks ago when he suffered from nasal congestion. He paid a visit to the office of a local medical doctor. The symptoms did not improve despite medication, so he was referred to the otolaryngology outpatient clinic. Endoscopic examination of the nose showed swelling of the bilateral anterior septum (Figure 1A,B), and fluctuation of the swelling was noted as well.

Abscess formation in the anterior nasal septum was impressed. Aspiration of the swollen area yielded 3 mL of dark red fluid, which was sent for aerobic and anaerobic bacterial cultures. After aspiration, Augmentin was administered, resulting in a dramatic improvement in nasal congestion. However, nasal congestion recurred 2–3 days after the procedure, and the results of aerobic and anaerobic cultures both showed no growth of bacteria. Since the nasal congestion and anterior nasal septum swelling did not subside, computed tomography (CT) scan was arranged. CT imaging showed presumed focal abscess formation (1.87 × 2.62 cm^2^) at the anterior nasal septum, near the anterior nasal nostril, and bilateral chronic maxillary sinusitis with calcification inside the left maxillary sinus (Figure 2A–C).

The patient had stage III right side buccal cancer and underwent radical surgery for the right buccal tumor and right supraomohyoid neck dissection 14 years ago. The pathological stage of the tumor was stage III, T3N0M0. Postoperative adjuvant radiotherapy was conducted (6200 cGy in 31 fractions on the right buccal tumor bed and right neck level I and II; 4600 cGy in 23 fractions on the left neck level I and II, and right whole neck except level I and II; seven fields, intensity-modulated radiotherapy, beam energy: 6 MV/photon) and the post-treatment follow-up course was uneventful, and no recurrence of the buccal cancer has been found to date. The anterior nasal septum was within the radiation therapeutic area, receiving a dosage of 2780 cGy according to the isodose curves from the Varian Eclipse treatment planning system. Moreover, the patient was found to have diabetes mellitus (DM) in 2009, when the glycated hemoglobin (HbA1c) was 10.6%. The patient’s DM has been well controlled medically since then with a prescription of half a tablet of Glimet (Glimepiride 2 mg + Metformin 500 mg) twice daily and one tablet of empagliflozin (10 mg) daily. The most recent laboratory test for HbA1c, conducted three weeks before the patient experienced nasal obstruction, showed a level of 6.5%. The patient also has hypertension and mixed hyperlipidemia under medical treatment with valsartan (160 mg) 1 tablet daily, and Vytorin 10/20 (Ezetimibe 10 mg + Simvastatin 20 mg) 1 tablet before sleep orally.

A preoperative complete blood cell count, conducted one day before surgery, revealed leukopenia with lower absolute neutrophil count (ANC) with a white blood cell 3000/µL (normal range 3900–10,600/µL) and ANC 1530/µL (normal range 1800–7800/µL). On the day of surgery, the blood cell count was checked again, revealing an even lower WBC count of 2000/µL and ANC of 1340/µL. The patient underwent incision and drainage of the anterior nasal septum abscess and bilateral functional endoscopic sinus surgery under general anesthesia. Initially, the septal abscess was drained. Discharge from the anterior nasal septum was collected, and mycobacterial smear and culture, as well as aerobic, anaerobic bacterial cultures, and fungal culture, were performed. Localized debridement of necrotic tissue and thorough abscess drainage were conducted. Subsequently, bilateral endoscopic sinus surgery was performed. Intraoperatively some compact, yellowish, clay-like material was found inside the left side maxillary sinus (Figure 3). All the material was removed and the sinus mucosa of the maxillary sinus was smooth without any erosion or destruction.

The culture of the nasal septum abscess was positive for *Aspergillus flavus*, with no growth on the mycobacterial and bacterial cultures. Pathological examinations showed the presence of fungal hyphae and special stains, including PAS (periodic acid–Schiff) stain (Figure 4) and GMS (Grocott methenamine silver) stain (Figure 5), highlighted the fungal structure with acute angle and septate hyphae, consistent with *Aspergillus* infection. There was no fungal invasion pathologically. After surgery, the patient continued on the antibiotic for one week and received a one-month course of voriconazole (200 mg/tab), taken orally twice daily. During follow-up, the patient recovered from the infection and fungal ball sinusitis without recurrence and was doing well. The swelling of the anterior nasal septum resolved, and no more swelling was noted (Figure 6). He has been followed up for 6 months.

## 3. Discussion

Anterior nasal septum abscess is not rare, but a fungal abscess in the anterior nasal septum is not common. In addition to the fact that the cause of abscess was from *Aspergillus*, the fact that this patient also had a concomitant fungal ball in the maxillary sinus is worthy of further exploration. In two articles studying fungal balls in the paranasal sinuses of 74 and 181 patients, respectively, the research results showed that 22.6% and 10.5% of patients were asymptomatic [16,17]. In the other retrospective study of 48 patients specifically on the maxillary sinus fungal ball, 41.6% of cases were asymptomatic [18]. An anterior nasal septum abscess can cause nasal obstruction, which is the most common presentation [10,19], and patients can easily feel the symptom and seek medical treatment. In this case, maxillary sinus mycetoma, or fungal ball, could be asymptomatic and seemed to be an incidental finding on a CT scan because the patient did not have other nose discomfort except nasal obstruction, and the nasal obstruction was dramatically relieved right after the first-time aspiration of the nasal septum abscess. It can be inferred that the nasal obstruction was primarily due to the anterior nasal septum abscess and his left maxillary fungal ball is likely asymptomatic.

Fungal infection of the anterior nasal septum can potentially originate from a hematogenous (blood-borne) route. Infections such as mucormycosis and invasive fungal sinusitis can spread through the bloodstream, reaching the nasal septum and producing infection [13]. There is blood flow between the maxillary sinus and the anterior nasal septum. The blood supply to the anterior septum includes arteries such as the superior labial, anterior ethmoidal, greater palatine, and sphenopalatine arteries, which contribute to the vascular network supplying the nasal septum [20,21]. The maxillary sinus receives its blood supply from branches of the maxillary artery, including the posterior superior alveolar artery, which contributes to the arterial architecture of the sinus [20,22]. While there is not a direct connection between the blood vessels of the maxillary sinus and anterior nasal septum, they share arterial branches that contribute to the overall blood circulation within the nasal cavity. Additionally, nasal septal infection can also result from the progression of a fungal infection from the adjacent sinuses, where the fungus may reach the nasal septum via blood dissemination [6]. While hematogenous spread is one possible route for fungal infection of the anterior nasal septum, it can also occur through direct contact with contaminated hands, from adjacent skin, or by extension from neighboring structures [12]. These alternative routes highlight the potential for fungal infection to affect the anterior nasal septum through various mechanisms beyond hematogenous dissemination. Several reported cases, which showed localized infections involving only the nasal septum, were described as a complication of sinusitis, recent trauma or dental procedures [19,23,24]. In the present case, there was no wound on the skin of the nostrils or the nasal mucosa and the patient did not have dental procedures or trauma before the occurrence of nasal obstruction. This indicates that the fungal infection in the anterior nasal septum most likely spread from the sinuses to the nasal septum through the hematogenous route, leading to abscess formation and tissue involvement.

Is it possible that the causal relationship between the fungal infections in the two locations (anterior septum and maxillary sinus) was reversed? In other word, did the mycetoma in the maxillary sinus come from the anterior septum fungal abscess? The anterior nasal septum space is a tiny little space made tightly adhesive by the mucoperichondium, nasal septum cartilage or bone and overlying nasal mucosa. This space is sealed off and usually compact and does not allow any fluid collections to be contained under normal circumstances [25]. The maxillary sinus is an air-filled space, and it communicates with the nasal cavity through an ostium. Compared with the sealed-off space of the anterior nasal septum, the probability of fungal infection in the maxillary sinus is much higher. The common routes of fungal ball in sinuses include nasal passages, damage or irritation to the nasal mucosa, sinus drainage pathways and pre-existing sinus conditions like allergic fungal sinusitis [26,27,28]. A fungal abscess of the anterior nasal septum has not been reported as an etiology of maxillary sinus mycetoma before. Considering the anatomical structures and clinical characteristics of the fungal infection, a fungal abscess of the nasal septum is less likely to be the cause of maxillary sinus mycetoma in this case.

Irradiation from radiotherapy on human body tissue can potentially make the tissue more susceptible to infection. Radiation therapy damages cancer cells, but it can also harm healthy cells in the treatment area, including those involved in the immune response. This damage to healthy tissue, particularly the suppression of the immune system, can increase the risk of infection [29]. Additionally, radiation-induced wound infections have been reported, especially in operated soft tissues, due to factors such as increased risk of infection and resistance to standard treatment modalities [30]. Furthermore, radiation therapy can cause low white blood cell counts, increasing susceptibility to infections [31]. In the present case, the anterior nasal septum fell within the scope of the irradiation during the course of postoperative adjuvant radiation for the oral cancer 14 years ago. Although it was not at the center of the irradiation, it still withstood a dose of 2780 cGy. It is possible that this exposure became a potential risk factor for infection and made the tissue more susceptible to infection years later.

Studies show that the most common organisms detected in the culture of nasal septal abscess are *Staphylococcus aureus*, *Staphylococcus epidermidis*, *Streptococcus pneumoniae*, beta-hemolytic group A *Streptococcus*, *Haemophilus influenzae*, and anaerobic bacteria [10,32]. If abscess fluid can be obtained at the beginning, bacterial, fungal, and mycobacterial cultures should be performed with prescribing empiric antibiotics covering a broad spectrum of potential pathogens commonly associated with this condition. If the empiric antibiotics fail, fungal infection or other less common etiologies should be taken into account, and the laboratory tests or pathological examinations play a crucial role in establishing the diagnosis [1,3,4,6,12,13,14,15]. Pathological examination of tissue samples through histology can also confirm fungal infections by identifying fungal elements like hyphae or spores within the tissue. This method provides direct visualization of fungal structures and can be especially useful when the clinicians do not cultivate the abscess fluid or cultures are negative or not feasible [33]. If the clinicians neglected to perform a fungal culture during the initial needle aspiration or incision and drainage of the abscess, we recommend conducting a fungal culture and pathological examination during the subsequent procedure. Symptoms of nasal septum abscess usually are recurrent because truly effective antifungal agents have not been used.

While most human *Aspergillus* infections are pulmonary and caused by the species *Aspergillus fumigatus*, *Aspergillus flavus* is more commonly found in sinus isolates [34,35]. *Aspergillus flavus* is unique to its thermotolerant character, and it can survive at temperatures that other fungi cannot [36]. *Aspergillus flavus* has a minimum growth temperature of 12 °C and a maximum growth temperature of 48 °C; the optimum growth temperature is 37 °C [37], which happens to be the human body temperature and explains why this saprophyte causes infections in human tissue. The treatment of mycetoma, or fungal ball, of the maxillary sinus is straightforward. Endoscopic sinus surgery remains the standard approach; the decision to use external surgical techniques should be individualized based on the patient’s condition and the extent of the fungal ball [18]. As for the fungal abscess of the anterior nasal septum, incision and drainage of the abscess with or without debridement of the adjacent tissue is the treatment of choice. In addition, another important point is to include this disease on the list of different diagnoses and remember to conduct a fungal culture or pathological examination in the subsequent incision and drainage of the abscess. Antifungal medication is not usually prescribed as the first-line treatment because of the uncommon incidence of a fungal abscess of the anterior nasal septum and the potential hazardous side effect of liver toxicity [38]. The implementation of a fungal culture or pathological examination is the basis for administering antifungal medication. Polymerase chain reaction of the ribosomal internal transcribed spacers (ITS) with the fungus-specific primers ITS1f and ITS5 was reported to successfully identify the fungal pathogens when fungal culture and pathology could not confirm the diagnosis [1,5].

There are 11 case reports of fungal infection or abscess of the anterior nasal septum in the literature [1,2,3,4,5,6,7,8,9,10,11]. Several pathogenic fungal species, including *Aspergillus flavus*, *Aspergillus thermomutatus*, *Fusarium verticillioides*, *Scedosporium apiospermum*, were detected by various methods. Immunodeficiency or immunocompromised status were noted in 10 out of the 11 patients (90.9%), indicating that this disease is more likely to occur in immunocompromised patients (Table 1). In our presented case, leukopenia was found with the fungal abscess of the nasal septum, which showed the patient was not immunocompetent. Previous adjuvant radiotherapy for oral cancer and diabetes mellitus were contributory factors to the immunological status of this patient. Since most patients with a fungal abscess of the anterior nasal septum were in an immunocompromised status, rapid establishment of the diagnosis and prompt and appropriate treatment to prevent rapid progression of the infection from deterioration into septicemia or central nervous system infection is particularly important. Severe complications, such as saddle nose, ablepsia and mortality, were reported [1,3,5]. Three cases of the 11 published fungal abscesses of the anterior nasal septum were found to be related to sinusitis. Two of them occurred several weeks and two weeks after endoscopic sinus surgery for the maxillary sinus and sphenoid sinus fungal ball, respectively [2,6]. In the one remaining case, the fungal abscess of the anterior nasal septum was concomitant with the maxillary fungal sinusitis [1]. In our case, the fungal abscess of the nasal septum was concomitant with the maxillary sinusitis, which is similar to the latter case. Beta-D-Glucan (BDG) is a fungal-cell-wall polysaccharide released into the bloodstream of patients with invasive candidiasis, invasive aspergillosis and other invasive fungal infections, except invasive zygomycosis, cryptococcosis, and *Blastomyces dermatitidis* [39]. According to the clinical manifestations and imaging studies in the present case, an invasive fungal infection was not likely, so BDG was not tested. The postoperative pathological examination also showed no fungal invasion. Among the 11 reported cases of fungal abscess of the anterior nasal septum, 10 were invasive fungal infections (90.9%). This is consistent with the fact that most of these patients were immunocompromised, as mentioned above. Perhaps because the case numbers of fungal abscesses of the anterior nasal septum are low, BDG has not been tested in any of the patients. Unlike some diagnostic tests that are specific to a single pathogen, the BDG assay is useful for detecting a wide range of invasive fungal infections [40]. It is a non-invasive diagnostic alternative that requires only small sample volumes and can provide a faster turnaround time compared to traditional culture-based tests [41], making it an important tool for assisting in early diagnosis.

## 4. Conclusions

We presented a rare case of concomitant fungal abscess of the anterior nasal septum and maxillary sinus mycetoma caused by *Aspergillus flavus*. The result of fungal culture and pathological examination enabled us to deduce that the fungal abscess of the anterior nasal septum came from the left maxillary sinus fungal ball. When an abscess of the anterior nasal septum is found clinically or on a CT scan, surgical treatments, including needle aspiration, incision and drainage, or debridement of the affected tissue, can be performed without difficulty because of the anatomical location of the disease. If empiric antibiotics are not effective initially, fungal culture and pathological examination should be performed. Patients with a fungal abscess of the anterior nasal septum often present in an immunocompromised state. Prompt establishment of a diagnosis and administration of antifungal agents are crucial for successful treatment.

## Figures and Tables

**Figure 1 jof-10-00497-f001:**
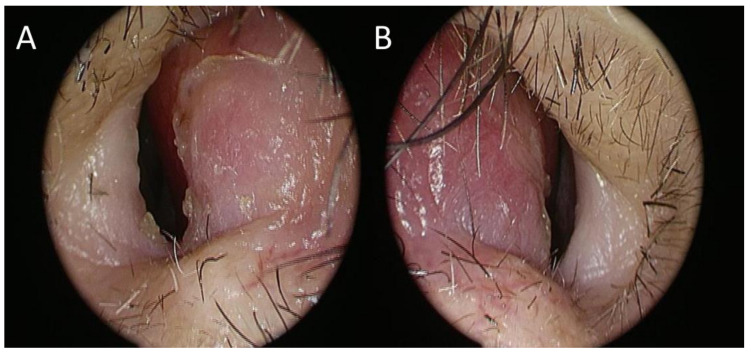
Swelling of the bilateral anterior nasal septum; right side (**A**), left side (**B**).

**Figure 2 jof-10-00497-f002:**
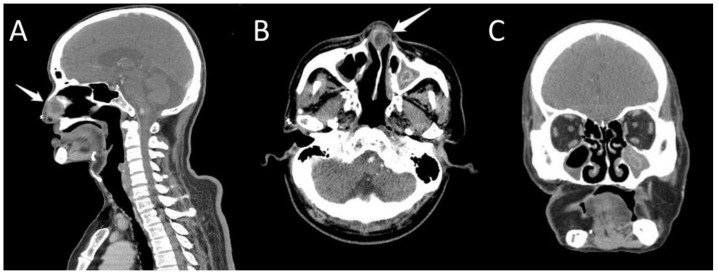
Computed tomography showed lower density in the anterior nasal septum (arrow) in the sagittal (**A**) and axial view (**B**) and calcification in the left side maxillary sinus in the axial (**B**) and coronal view (**C**).

**Figure 3 jof-10-00497-f003:**
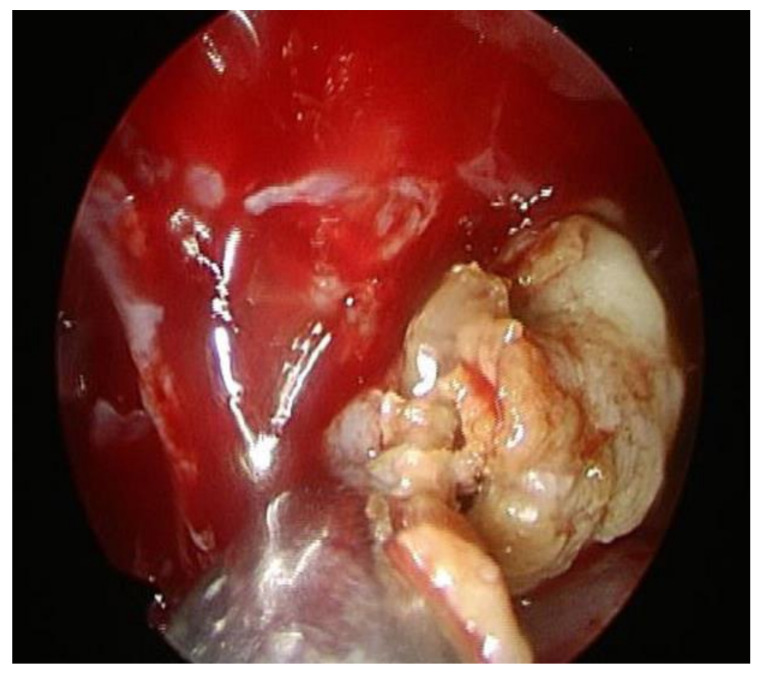
Compact, yellowish, clay-like material was found inside the left side maxillary sinus.

**Figure 4 jof-10-00497-f004:**
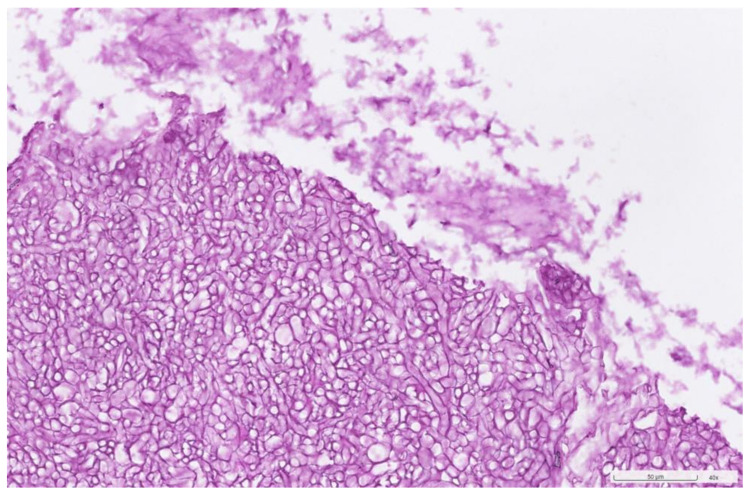
Periodic acid–Schiff (PAS) stain showed magenta and dense accumulation of septate hyphae with acute-angle branching from the yellowish clay-like material in the left maxillary sinus (original magnification ×40).

**Figure 5 jof-10-00497-f005:**
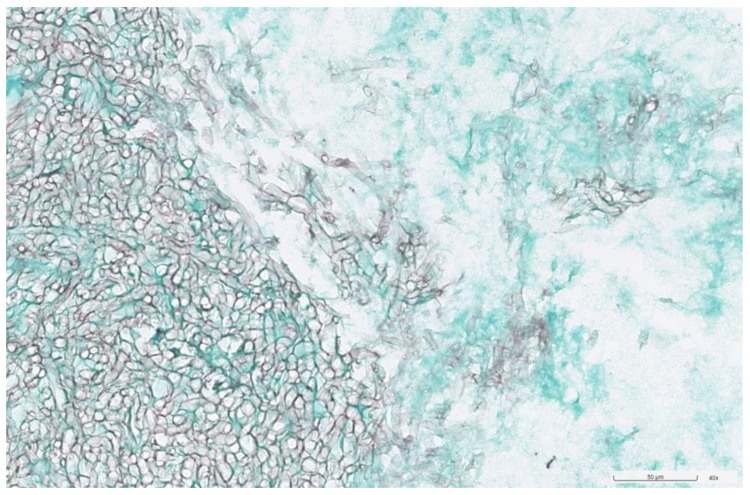
GMS (Grocott methenamine silver) stain showed darkly stained septate fungal hyphae with branching (original magnification ×40).

**Figure 6 jof-10-00497-f006:**
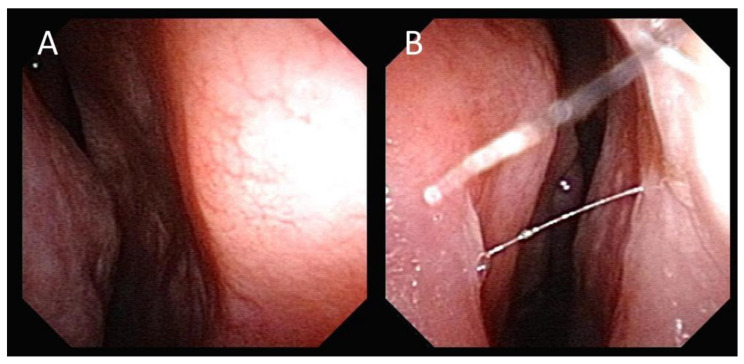
Three months after surgical drainage and antifungal treatment, the bilateral anterior nasal septum swelling was in a complete resolution; right side (**A**), left side (**B**).

**Table 1 jof-10-00497-t001:** Clinical characteristics and demographics of reported cases of fungal abscesses of the anterior nasal septum in the English literature.

No.	Author(s), Publication Date	Age	Gender	Fungus	Method of Detection of Fungus	Invasive Fungal Infection	Patient’s Immune Status
1	Siberry et al., 1997 [9]	15	Male	*Aspergillus*	Fungal culture and pathology	Yes	Immuno-compromised
2	Dornbusch et al., 2005 [5]	9	Male	*Fusarium verticillioides*	PCR ‡ of the ribosomal ITS §	Yes	Immuno-compromised
3	Walker et al., 2007 [4]	64	Male	*Aspergillus flavus*	Fungal culture and pathology	No	Immuno-compromised
4	Debnam et al., 2007 [10]	17	Male	*Aspergillus flavus*	Fungal culture and pathology	Yes	Immuno-compromised
5	Naeem et al., 2011 [8]	5	Male	*Aspergillus flavus*	Fungal culture and pathology	Yes	Immuno-compromised
6	Cho et al., 2012 [7]	52	Male	*Aspergillus* species	Pathology	Yes	Immuno-compromised
7	Patel et al., 2015 [2]	51	Female	*Scedosporium apiospermum* (nasal septum) †	Fungal culture (nasal septum) †	Yes	Immuno-competent
8	Kishimoto et al., 2017 [1]	59	Male	*Scedosporium apiospermum* in both nasal septum and left side maxillary sinus	PCR ‡ of the ribosomal ITS § (nasal septum and maxillary sinus); pathology (maxillary sinus)	Yes	Immuno-compromised
9	Sohn et al., 2018 [6]	79	Male	*Aspergillus* species	Pathology (maxillary sinus, anterior nasal spetum)	Yes	Immuno-compromised
10	Parent-Michaud et al., 2019 [11]	NA *	NA *	*Aspergillus thermomutatus*	Draft genome sequence	Yes	Immuno-compromised
11	Rochdi et al., 2020 [3]	53	Female	*Aspergillus flavus*	Fungal culture	Yes	Immuno-compromised
12	The present case	57	Male	*Aspergillus flavus* (nasal septum); *Aspergillus* spp. (maxillary sinus)	Fungal culture (nasal septum); pathology (maxillary sinus)	No	Immuno-compromised

Abbreviations, * NA: not available; ‡ PCR: Polymerase chain reaction; § ITS: internal transcribed spacers. † Species of fungi in maxillary sinus not mentioned. Detection method of maxillary fungal ball not mentioned.

## Data Availability

The original contributions presented in the study are included in the article, further inquiries can be directed to the corresponding author.

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
