# Peer review of "Fungal Abscess of Anterior Nasal Septum Complicating Maxillary Sinus Fungal Ball Rhinosinusitis Caused by Aspergillus flavus: Case Report and Review of Literature"

_jof, 2024, doi:10.3390/jof10070497_

Round 1

Reviewer 1 Report

Comments and Suggestions for Authors

YanG, Luo, and Cheng submit a case report and review of the literature on fungal abscess of the anterior nasal septum complicating maxillary sinus fungal ball rhinosinusitis caused by Aspergillus flavus.

Major comments:

-          The authors may want to add a sentence that while overall human Aspergillus infections are pulmonary and caused by the species fumigatus, it is Aspergillus flavus that is more common among sinus isolates. The authors could cite a general review of Aspergillus infection which supports this.

Minor comments:

-          Use of the word Aspergillus or flavus should be in italics, as they are Latin genus and species names.

-          References 1, 3, 5, 8, 11, 15, and 35: the titles of these references have microbiologic organisms listed with Latin genus and species names, and these Latin words should be in italics.

Comments on the Quality of English Language

This paper should be reviewed by a native English speaker. Please try to use less of the passive voice with your next revision of this manuscript.

Reviewer 2 Report

Comments and Suggestions for Authors

This case report with reviewing septal fungal infection is well written and informative for clinical management of fungal infection.  The shortage of this report, comparing invasive type of fungal infection in sinonasal cavity with bad prognosis.  The readers would like to know the serum beta-D-glucan level, which we are missing, in comparison with invasive type.  Discussion is well done with each point, but authors should distinguish or not this case from so called invasive type of aspergillosis to nasal septum, not to orbital or intracranial space, even with mycetoma in left maxillary sinus cavity.

 So, I think that with adding some comments on these questions, this paper can be accepted for publication.

Round 2

Reviewer 1 Report

Comments and Suggestions for Authors

The authors have done a great job with improvement in English language.

- This sentence is still troublesome:  "Initially, the septum abscess was drained, yielding first performed. Some discharge from the anterior nasal septum, mycobacterial smear and culture, aerobic and anaerobic bacterial cultures, and fungal culture were performed".

- The authors have added reference 35 as a generalized review of aspergillosis that supports flavus as a species involved in the sinuses, but it is from 1989. Could also supplement with PMID 34644473 from 2021.

Comments on the Quality of English Language

Much improved; still some minor change needed per comments to the authors.
